# Deep Phenotyping of Superficial Epidermolytic Ichthyosis due to a Recurrent Mutation in *KRT2*

**DOI:** 10.3390/ijms23147791

**Published:** 2022-07-14

**Authors:** Yuika Suzuki, Takuya Takeichi, Kana Tanahashi, Yoshinao Muro, Yasushi Suga, Tomoo Ogi, Masashi Akiyama

**Affiliations:** 1Department of Dermatology, Nagoya University Graduate School of Medicine, 65 Tsurumai-cho, Showa-ku, Nagoya 466-8550, Japan; tanahashi@med.nagoya-u.ac.jp (K.T.); ymuro@med.nagoya-u.ac.jp (Y.M.); makiyama@med.nagoya-u.ac.jp (M.A.); 2Department of Dermatology, Juntendo Urayasu Hospital, 2-1-1 Tomioka, Urayasu 279-0021, Japan; ysuga@juntendo.ac.jp; 3Department of Genetics, Research Institute of Environmental Medicine (RIeM), Nagoya University, Furo-cho, Chikusa-ku, Nagoya 464-8601, Japan; togi@riem.nagoya-u.ac.jp; 4Department of Human Genetics and Molecular Biology, Nagoya University Graduate School of Medicine, 65 Tsurumai-cho, Showa-ku, Nagoya 466-8550, Japan

**Keywords:** genodermatosis, ichthyosis bullosa of Siemens, keratin 2, keratinopathic ichthyoses, superficial epidermolytic ichthyosis

## Abstract

Superficial epidermolytic ichthyosis (SEI) is an autosomal dominant inherited ichthyosis. SEI is caused by mutations in *KRT2* and frequently shows erythroderma and widespread blistering at birth. We report the clinical manifestations of two patients from a Japanese family with SEI caused by a hotspot mutation, p.Glu487Lys, in *KRT2*. In addition, we summarize previous reports on SEI patients with the identical mutation. One of the two patients had disease onset at the age of 7 months. The other patient’s age of onset is unknown, but it was in childhood. Neither of the two patients showed erythroderma. To perform deep phenotyping, we studied the age of onset and the frequency of erythroderma in 34 reported SEI cases with the p.Glu487Lys mutation, including the present cases. Among the cases with sufficient clinical information, 44.4% of the cases that were due to p.Glu487Lys in *KRT2* occurred at birth. Erythroderma was observed in 11.1% of the cases with p.Glu487Lys in *KRT2*.

## 1. Introduction

Superficial epidermolytic ichthyosis (SEI) is an autosomal dominant inherited skin disease caused by mutations in the keratin 2 gene (*KRT2*). It is characterized by mild epidermal hyperkeratosis of the extremities, very shallow blister formation and superficial denuded areas of the hyperkeratotic epidermis [1]. A large number of SEI patients have been reported to have causative mutations at the 487th glutamic acid in the helix termination motifs of keratin 2. Thus, the glutamate is known to be a mutational hotspot [2]. In this article, we report a Japanese family with SEI caused by the recurrent mutation p.Glu487Lys in *KRT2*. Moreover, we describe clinical, histopathological, and molecular genetic findings in SEI patients in the family and provide a brief review of the literature on cases with the mutation p.Glu487Lys in *KRT2*.

## 2. Patients, Materials and Methods

### 2.1. Case History

#### 2.1.1. Patient 1 (the Proband)

A 40-year-old man from an unrelated Japanese family (the proband in the pedigree, Appendix A) was referred to our dermatology clinic with a history of dry skin and blistering since childhood (specific age unknown). The blisters and erosions were seen until about 12 years of age but they rarely occurred after that age. Even in adulthood, the stratum corneum peeled off easily when tape was applied to the skin, but he was usually suffering from only mild dryness. It is unclear whether he had erythroderma at birth, but he was never aware of erythroderma. His parents and siblings had no similar skin disorders.

#### 2.1.2. Patient 2

The 19-month-old son of the proband (Patient 1) (Appendix A) had a history of recurrent blisters and erosions of the skin since the age of 7 months. The frequency of the symptoms decreased with growth. No abnormalities in growth or development were noted. Patient 2’s mother and father (Patient 1) were unrelated and he had no siblings.

### 2.2. Histopathological Examination

A punch biopsy was taken from the skin of the anterior lower leg of Patient 1.

### 2.3. Genetic Testing

Genomic DNA from the peripheral blood mononuclear cells of Patient 1 was used for whole-exome sequencing analysis and Sanger sequencing analysis. Genomic DNA extracted from the saliva of Patient 2 was used for Sanger sequencing analysis. Whole-exome sequencing was conducted. The data were analyzed using Sequence Analysis Software (CLC Main Workbench, Filgen Incorporated, Japan). The transcript ENST00000309680.4 was used as a reference for *KRT2*. The *KRT2* mutation identified by the whole-exome sequencing was verified by Sanger sequencing.

### 2.4. Literature Reveiw

A literature review was conducted in PubMed (June 1986 to March 2022) using the terms “superficial epidermolytic ichthyosis” or “ichthyosis bullosa of Siemens”. Genetic forms, ages of onset, and presence/absence of erythroderma in SEI patients were examined. Although genotype–phenotype correlations in SEI have not been clarified [3], given the possibility of genotype–phenotype correlations, the present review only includes SEI patients with the *KRT2* mutation p.Glu487Lys, identical to the mutation found in the present family.

## 3. Results

### 3.1. Clinical Presentations

#### 3.1.1. Patient 1 (the Proband)

On physical examination, he presented with dark, hyperkeratotic skin with scaling on the elbows and the lower extremities (Figure 1A–C). No fresh blisters nor erosions were present.

#### 3.1.2. Patient 2

Physical examination revealed light gray, hyperkeratotic skin at the joints of the extremities (Figure 1D). Superficial denuded areas were also observed where dressings had been applied (Figure 1E). No fresh blisters nor erosions were present. 

### 3.2. Histopathological Examination

A skin biopsy specimen from the anterior lower leg of Patient 1 showed marked hyperkeratosis. Perinuclear vacuoles and basophilic, irregularly sized keratohyaline granules were present within the cytoplasm of keratinocytes from the upper spinous and granular layers of the epidermis (Figure 1F). 

### 3.3. Genetic Testing

Genomic DNA from the peripheral blood leucocytes of Patient 1 was used for whole-exome sequencing analysis. The data revealed a heterozygous missense mutation in *KRT2*, c.1459G>A (p.Glu487Lys), which was confirmed by Sanger sequencing (Figure 1G). No other potentially pathogenic mutations were identified in *KRT2*, nor in any other gene associated with congenital ichthyoses. Subsequently, genomic DNA from the saliva of Patient 2 was used for Sanger sequencing, and the identical heterozygous missense mutation was identified in Patient 2.

### 3.4. Literature Review

SEI was first reported by Siemens in 1937 as a mild form of epidermolytic ichthyosis. As far as we investigated, 20 families with the mutation p.Glu487Lys have been reported, and clinical information has been described in 34 cases, including the present ones (Table 1) [1,2,4,5,6,7,8,9,10,11,12,13,14,15,16,17]. Among them, the age of onset and the presence/absence of erythroderma were reported in 27 cases each. In total, 12 of these 27 cases (44.4%) had visible skin symptoms, such as erythroderma, hyperkeratosis, or blistering, when they were born. Three cases occurred at 1 month of age, four cases by 3 months of age, two cases at 4 months of age, two cases at 6 months of age, one case at 7 months of age, one case at 8 months of age, and two cases at 18 months of age. Erythroderma was observed in only 3 of the 27 cases (11.1%). Significantly, there was intrafamilial clinical heterogeneity in the affected family members.

## 4. Discussion

SEI was previously called ichthyosis bullosa of Siemens and was renamed to its current name at the First Ichthyosis Consensus Conference in Sorèze in 2009 [18]. SEI presents clinical features similar to those of epidermolytic ichthyosis, but the clinical symptoms of SEI are milder, without hyperkeratosis of the palms and soles, and they generally improve with age [17]. 

SEI-causing gene mutations have been identified mostly in the helix initiation and termination motifs of *KRT2*. The mutations in these helix boundary motifs of keratins affect the assembly of an intermediate filament network more than the mutations occurring at other sites of keratin molecules, which impair the stability of the protein [2,5]. The present variant in *KRT2* is predicted to be a disease-causing variant by computational (in silico) predictive programs [19,20,21]. The major epidermal keratins are keratins 9, 10, 14, and 16 for the type I class and keratins 1, 2, 5, and 6 for the type II class. The expression of each keratin is specific to locations and differentiation stages of epidermal keratinocytes [22]. In keratinocytes of the upper spinous to granular layers, *KRT2* expression is upregulated. Thus, in SEI, histopathological findings are characterized by granular degeneration consistent with the site of *KRT2* expression [1]. Consistent with the findings, the skin of SEI patients is usually fragile and the outer layers of the epidermis have the tendency to peel off, producing localized superficial denuded areas. This characteristic clinical feature is called “the Mauserung phenomenon”, or molting [5]. Patient 2 had this symptom (Figure 1E). 

SEI was first reported by Siemens in 1937, and the absence of erythroderma was proposed as part of the definition of SEI [23]. However, it has become clear that there are cases of SEI with erythroderma. Thus, at the conference in 2009, erythroderma was added to the list of initial symptoms of SEI [18]. Furthermore, the onset of SEI was considered to be usually at birth. However, no clear statistical data were reported. A search of “The Human Gene Mutation Database” revealed a missense mutation in the 487th glutamate in the helix termination motif of *KRT2* to be the most frequently reported among diverse ethnicities and geographic areas, suggesting that the 487th glutamic acid is a mutational hotspot [24]. Thus, we performed deep phenotyping for the age of onset and the presence/absence of erythroderma in the reported SEI cases with the identical *KRT2* mutation p.Glu487Lys.

The same genotype has been reported in 22 families, and clinical information has been described in 34 cases in 20 families, including the present patients (Table 1) [1,2,4,5,6,7,8,9,10,11,12,13,14,15,16,17]. Among them, the age of onset and the presence/absence of erythroderma have been reported in 27 cases each. In total, 12 of the 27 cases (44.4%) had the disease at birth. Three cases occurred at 1 month of age, four cases by 3 months of age, two cases at 4 months of age, two cases at 6 months of age, one case at 7 months of age, one case at 8 months of age, and two cases at 18 months of age. Erythroderma was observed in only 3 out of the 27 cases (11.1%) (Figure 2). Significantly, there was intrafamilial clinical heterogeneity in the affected family members. Generally, the number of cases with erythroderma in SEI is small and it was confirmed that the majority of patients with the recurrent *KRT2* mutation p.Glu487Lys never showed erythroderma. Moreover, it was found that about half of the patients show the disease at birth and the other half after birth.

## 5. Conclusions

Our findings indicate that it was assumed to be difficult to distinguish SEI from epidermolytic ichthyosis only from the presence/absence of erythroderma. When the possibility of epidermolytic ichthyosis cannot be excluded based on the patients’ clinical features, then genetic diagnosis of SEI is important for definite diagnosis and genetic counseling, including the prediction of symptoms and prognosis. We also demonstrated that children at a high risk of SEI from their family history, especially of SEI due to the *KRT2* mutation p.Glu487Lys, should be carefully followed up until about 18 months of age, even if they have no symptoms immediately after birth.

## Figures and Tables

**Figure 1 ijms-23-07791-f001:**
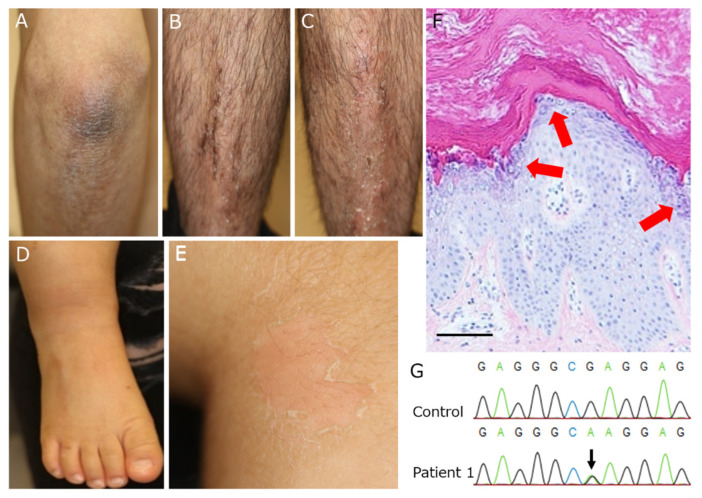
Clinical and histopathological features of the present patients and the causative *KRT2* mutation. (**A**–**C**) In Patient 1, brown, hyperkeratotic lesions with scaling are observed on the elbow (**A**) and the lower legs (**B**,**C**). (**D**) In Patient 2, light gray, hyperkeratotic lesions are seen on the extensor side of the ankle joint and the dorsa of the right foot. (**E**) A superficial denuded area, called “Mauserung phenomenon”, is also observed on the flexor side of the knee of Patient 2. (**F**) Histopathological observations of the hyperkeratotic skin on the anterior lower leg of Patient 1 reveal significant hyperkeratosis and granular degeneration restricted to the upper spinous and granular layers of the epidermis (scale bar: 50 μm). (**G**) Sanger sequencing confirms the heterozygous mutation within *KRT2*, c.1459G>A (p.Glu487Lys), in Patient 1.

**Figure 2 ijms-23-07791-f002:**
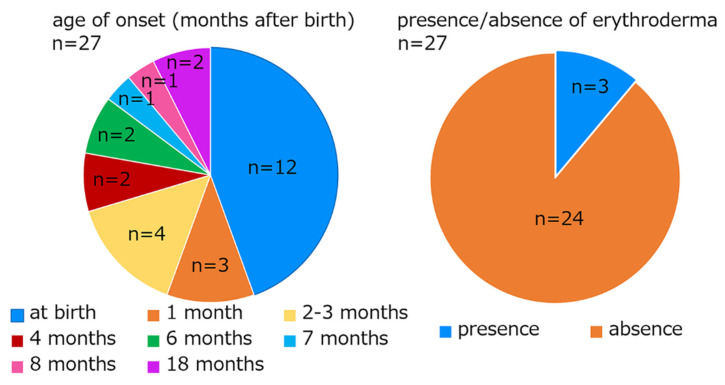
The age of onset and the presence/absence of erythroderma in SEI patients with the *KRT2* mutation p.Glu487Lys. Only the cases with clinical information are included in this figure.

**Table 1 ijms-23-07791-t001:** Review of published cases of superficial epidermolytic ichthyosis with the identical mutation p.Glu487Lys in *KRT2*.

Family-Case No.	Age/Sex	Familial or Sporadic	Age at Onset	Erythroderma Present/Absent	Reference
1-1	10 yrs./female	familial	within a few months	absent	McLean et al., 1994 [4]
2-1	—/female	sporadic	—	absent	McLean et al., 1994 [4]
3-1	26 yrs./male	familial	at birth	present	Rothnagel et al., 1994 [5]
4-1	6 yrs./female	familial	at birth	absent	Rothnagel et al., 1994 [5]; Traupe et al., 1986 [6]
4-2	31 yrs./male	familial (father of No. 4-1)	at birth	absent	Rothnagel et al., 1994 [5]; Traupe et al., 1986 [6]
5-1	—/female	sporadic	at birth	present	Rothnagel et al., 1994 [5]
6-1	32 yrs./female	familial	at birth	absent	Kremer et al., 1994 [7]; Vakilzadeh and Kolde, 1991 [8]
6-2	31 yrs./male	familial (brother of No. 8-1)	at birth	absent	Kremer et al., 1994 [7]; Vakilzadeh and Kolde, 1991 [8]
6-3	13 yrs./female	familial (daughter of No. 8-1)	at birth	absent	Kremer et al., 1994 [7]; Vakilzadeh and Kolde, 1991 [8]
6-4	6 yrs./male	familial (son of No. 8-1)	at birth	absent	Kremer et al., 1994 [7]; Vakilzadeh and Kolde, 1991 [8]
7-1	—/male	familial	—	absent	Kremer et al., 1994 [7]
7-2	—/female	familial (daughter of No. 9-1)	—	absent	Kremer et al., 1994 [7]
7-3	—/male	familial (son of No. 9-1)	—	absent	Kremer et al., 1994 [7]
7-4	—/female	familial (daughter of No. 9-1)	—	absent	Kremer et al., 1994 [7]
8-1	13 yrs./female	familial	6 months	absent	Jones et al., 1997 [9]; Mills and Marks, 1993 [10]
9-1	3 yrs./male	sporadic	100 days	—	Yang et al., 1998 [11]
10-1	1 yrs./female	familial	at birth	present	Basarab et al., 1999 [12]
10-2	11 yrs./male	familial (cousin of No. 12-1)	6 months	—	Basarab et al., 1999 [12]
10-3	9 yrs./female	familial (cousin of No. 12-1)	4 months	—	Basarab et al., 1999 [12]
11-1	—/male	familial	at birth	absent	Suga et al., 2000 [2]
12-1	56 yrs./female	familial	8 months	—	Akiyama et al., 2005 [1]
12-2	28 yrs./male	familial (son of No. 14-1)	3 months	—	Akiyama et al., 2005 [1]
13-1	3 yrs./male	familial	1 months	—	Akiyama et al., 2005 [1]
14-1	6 yrs./male	familial	at birth	absent	Langan et al., 2010 [13]
14-2	20 mo./male	familial (brother of No. 16-1)	at birth	absent	Langan et al., 2010 [13]
15-1	2 yrs./male	familial	3 months	—	Cervantes et al., 2013 [14]
16-1	18 mo./female	sporadic	1 month	absent	Gameiro et al., 2016 [15]
17-1	5 yrs./male	sporadic	40 days	absent	Li et al., 2020 [16]
18-1	5 yrs./male	sporadic	4 months	absent	Diociaiuti et al., 2020 [17]
19-1	43 yrs./male	familial	—	absent	Diociaiuti et al., 2020 [17]
19-2	7 yrs./male	familial (son of No. 21-1)	18 months	absent	Diociaiuti et al., 2020 [17]
19-3	7 yrs./male	familial (son of No. 21-1)	18 months	absent	Diociaiuti et al., 2020 [17]
20-1	40 yrs./male	familial	—	absent	The present report
20-2	19 mo./male	familial (son of No. 22-1)	7 months	absent	The present report

Abbreviations: yrs., years old; mo., months old; —, not described.

## Data Availability

Not applicable.

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
