# Peer review of "Deep Phenotyping of Superficial Epidermolytic Ichthyosis due to a Recurrent Mutation in KRT2"

_ijms, 2022, doi:10.3390/ijms23147791_

Round 1

Reviewer 1 Report

This is a rare disease and review of the literature, as well as case-presentation and genetic analysis is very usefull.

I have some minor comments:

Abstract: You write 'Among the cases with sufficient clinical information, 44.4% of the cases that were due to p.Glu487Lys in KRT2 occurred at birth..' Do you mean that there were diagnostic symptoms at birth?

Literature review: when you describe 'had the disease at birth', do you mean visible symptoms at birth? Or do you mean cases where ichthyosis may be acquired? It should be very clear (phenotype).

'Erytroderma was observed in 11.1%'. Do you mean at birth or when the first symptoms appeared?

Figure 1: Please add that squares are males and circles females. Especially since other ichthyoses may be X-bound.

I think '..improve with age..' is more correct than '...improve with growth..'

Author Response

Response to Reviewer 1 Comments

Point 1: Abstract: You write 'Among the cases with sufficient clinical information, 44.4% of the cases that were due to p.Glu487Lys in KRT2 occurred at birth..' Do you mean that there were diagnostic symptoms at birth?

Response 1: Thank you so much for this comment. Yes, 44.4% of the cases with p.Glu487Lys in KRT2 had at least one diagnostic symptom of SEI (erythroderma, hyperkeratosis, or blistering) at birth.

Point 2: Literature review: when you describe 'had the disease at birth', do you mean visible symptoms at birth? Or do you mean cases where ichthyosis may be acquired? It should be very clear (phenotype).

Response 2: We appreciate this significant comment and suggestion. The words “had the disease at birth” mean “had visible symptoms at birth”. It does not mean cases where the ichthyosis may have been acquired. According to the reviewer’s valuable suggestion, we have revised the corresponding text as follows.

Lines 118-119 of the main text:

“12 of these 27 cases (44.4%) had visible skin symptoms, such as erythroderma, hyperkeratosis, or blistering, when they were born.”

Point 3: 'Erythroderma was observed in 11.1%'. Do you mean at birth or when the first symptoms appeared?

Response 3: We are thankful for the referee’s significant query. If the erythroderma was observed at any time during the clinical course of SEI (not only at birth or when the first symptoms appeared), then we defined the patient as having erythroderma.

Point 4: Figure 1: Please add that squares are males and circles females. Especially since other ichthyoses may be X-bound.

Response 4: We are grateful for this helpful comment. We agree with Reviewer 1 on the importance of describing “squares are males and circles females”, to rule out other ichthyoses that might be X-bound. However, Reviewer 2 has suggested that we remove the pedigree tree from Figure 1 in order to increase the sizes of the clinical images. Based on the supportive comments of Reviewer 1 and another reviewer, we have removed the pedigree tree from Figure 1 and added it as Supplementary Material, Figure S1. We have added the following text to our latest version as a supplementary caption.

Lines 188-189 of Supplementary Materials:

“Figure S1: The pedigree of the present family. Squares are males and circles are females. Solid squares represent affected individuals (P, the proband).”

The reference numbers of the Supplementary Materials have been modified accordingly.

Lines 45, 53 of the main text: “Figure S1”

Line 191 of Supplementary Materials: “Figure S2”

Point 5: I think '..improve with age..' is more correct than '...improve with growth..'

Response 5: Thank you for the supportive advice. We have revised the corresponding words as below.

Lines 133-134 of the main text

“… improve with age [17].”

Reviewer 2 Report

This is a well presented (mini) case series of 2 patients with SEI having pathogenic variants in hotspot areas of KRT2. It makes for good reading to a general audiance.

General comments;

Figure: the pedigree is quite straight forward I do not think that it needs to be included. The space could be used to increase the size of the clinical photos.

Even though the pathogenic missense variant has been described before, a line or two on it’s effect (in silico predictors of) on the protein’s stability, structure and function may be merited.  

Author Response

Response to Reviewer 2 Comments

Point 1: Figure: the pedigree is quite straight forward I do not think that it needs to be included. The space could be used to increase the size of the clinical photos.

Response 1: Thank you so much for the helpful suggestion. We agree that the pedigree is quite straightforward. Therefore, we have removed the pedigree tree from Figure 1. However, Reviewer 1 suggested that we give more details on the pedigree tree, since other ichthyoses can have X-linked inheritance. Therefore, the pedigree tree has been included as a supplementary file.

The capital letters or numbers attached to each element of Figure 1 and supplementary Figures have been modified accordingly.

Lines 188-189 of Supplementary Materials:

“Figure S1: The pedigree of the present family. Squares are males and circles are females. Solid squares represent affected individuals (P, the proband).”

Line 191 of Supplementary Materials: “Figure S2”

Point 2: Even though the pathogenic missense variant has been described before, a line or two on its effect (in silico predictors of) on the protein’s stability, structure and function may be merited.

Response 2: We appreciate the instructive comment. We investigated the effects of the missense variant by using tools which predict the possible impact of an amino acid substitution or the protein, such as “MutationTaster”, “PolyPhen-2”, “Sorting Intolerant From Tolerant (SIFT)”. According to the reviewer’s advice, we have added the following text and cited some references.

Lines 136-140 of the main text:

“The mutations in these helix boundary motifs of keratins affect the assembly of an intermediate filament network more than mutations occurring at other sites of keratin molecules that impair the stability of the protein [2, 5]. The present variant in KRT2 is predicted to be a disease-causing variant by computational (in silico) predictive programs [19, 20, 21].”

The newly added references:

  1. Schwarz, J.M.; Cooper, D.N.; Schuelke, M.; Seelow, D.; MutationTaster2: mutation prediction for the deep-sequencing age. Nat Methods. 2014, 11, 361-362. doi:10.1038/nmeth.2890
  2. Adzhubei, I.A.; Schmidt, S.; Peshkin, L.; Ramensky, V.E.; Gerasimova, A.; Bork, P.; Kondrashov, A.S.; Sunyaev, S.R.; A method and server for predicting damaging missense mutations. Nat Methods. 2010, 7, 248-249. doi:10.1038/nmeth0410-248.
  3. Ng, P.C.; Henikoff S.; Predicting deleterious amino acid substitutions. Genome Res. 2001, 11, 863-874. doi:10.1101/gr.176601.
